Bioreactor virome metagenomics sequencing using DNA spike-ins

Cremers Geert
Gambelli Lavinia
van Alen Theo
van Niftrik Laura
Op den Camp Huub J.M. h.opdencamp@science.ru.nl
Department of Microbiology, Institute of Water and Wetland Research, Faculty of Science, Radboud University , Nijmegen , Netherlands
Loman Nicholas
Electronic publication date: 2018 Feb 7
Publication date: 2018
Volume: 6
Electronic Location ID: e4351
Received 2017 Apr 3; Accepted 2018 Jan 19
Copyright: ©2018 Cremers et al.
Copyright year: 2018
Copyright holder: Cremers et al.
License: This is an open access article distributed under the terms of the Creative Commons Attribution License, which permits unrestricted use, distribution, reproduction and adaptation in any medium and for any purpose provided that it is properly attributed. For attribution, the original author(s), title, publication source (PeerJ) and either DOI or URL of the article must be cited.
License URL: https://creativecommons.org/licenses/by/4.0/

Keywords: Metagenome, Metavirome, DNA spiking, Bacteriophage, Multiple displacement amplification, Virus

Funding: ERC Advanced grants ERC-339880 ERC-669371 Netherlands Organisation for Fundamental Research Gravitation SIAM024002002 This work was supported by ERC Advanced grants (ERC-339880 and ERC-669371) and by the Netherlands Organisation for Fundamental Research Gravitation grant (SIAM024002002). The funders had no role in study design, data collection and analysis, decision to publish, or preparation of the manuscript.

==============================
With the emergence of Next Generation Sequencing, major advances were made with regard to identifying viruses in natural environments. However, bioinformatical research on viruses is still limited because of the low amounts of viral DNA that can be obtained for analysis. To overcome this limitation, DNA is often amplified with multiple displacement amplification (MDA), which may cause an unavoidable bias. Here, we describe a case study in which the virome of a bioreactor is sequenced using Ion Torrent technology. DNA-spiking of samples is compared with MDA-amplified samples. DNA for spiking was obtained by amplifying a bacterial 16S rRNA gene. After sequencing, the 16S rRNA gene reads were removed by mapping to the Silva database. Three samples were tested, a whole genome from Enterobacteria P1 Phage and two viral metagenomes from an infected bioreactor. For one sample, the new DNA-spiking protocol was compared with the MDA technique. When MDA was applied, the overall GC content of the reads showed a bias towards lower GC%, indicating a change in composition of the DNA sample. Assemblies using all available reads from both MDA and the DNA-spiked samples resulted in six viral genomes. All six genomes could be almost completely retrieved (97.9%–100%) when mapping the reads from the DNA-spiked sample to those six genomes. In contrast, 6.3%–77.7% of three viral genomes was covered by reads obtained using the MDA amplification method and only three were nearly fully covered (97.4%–100%). This case study shows that DNA-spiking could be a simple and inexpensive alternative with very low bias for sequencing of metagenomes for which low amounts of DNA are available.

Introduction

Microbial research has been mainly culture-based since the work of Pasteur and Koch. This has led to great improvement of our knowledge of the microbial and viral world. However, our knowledge is probably still only the tip of the iceberg, as most of the microorganisms cannot be cultured (Rosario & Breitbart, 2011). In recent years, there has been a greater focus on the hidden bacterial and viral ‘black matter’ (Krishnamurthy & Wang, 2017) since the development of next generation sequencing (NGS) techniques which allow the determination of the microbial community without the need for cultivation. Without the necessity of cultivation prior to sequencing, organisms that cannot be cultured under artificial conditions are now being sequenced in increasing numbers. This is especially true for bacteria; sequencing viral black matter from environmental samples is still hampered by a variety of factors. Besides the obvious problem that not all viruses are DNA viruses (Steward et al., 2013), there is also the challenge of the low quantity of DNA that can be retrieved from viruses. Although viruses outnumber bacteria five to 25 times in numbers (Fuhrman, 1999; Clokie et al., 2011), the fact that viruses have on average a significantly smaller genome size means that the viral DNA yield from any given sample is significantly lower compared to bacterial DNA (approximately tens of attograms per virion vs femtograms per bacterium) (Brum & Sullivan, 2015). Because of this, DNA is often extracted using DNA extraction methods optimized for virions gathered from a large amount (20–200 L) of sample (Breitbart et al., 2002; Thurber et al., 2009; Duhaime et al., 2012; Steward et al., 2013).

Since sampling large amounts is not always possible and because there is loss of DNA in every step of DNA isolation protocols, the final yield of viral DNA remains low. Therefore, two options are available for sequencing. (A) Sequence very low quantities (femtograms to picograms of DNA). However, this method is sensitive to contamination and might skew community composition. It also complicates binning and assembly afterwards because of the lack of template DNA (Rinke et al., 2016). (B) Amplify the low amount of DNA before sequencing. Several methods are available for this purpose (Duhaime et al., 2012; Brum & Sullivan, 2015), with the Multiple Displacement Amplification (MDA) method being the most widely used. However, MDA has a few drawbacks. Because it is amplification, it unavoidably introduces a bias (Kim & Bae, 2011; Marine et al., 2014) and furthermore information of relative abundance in the original sample is lost (Yilmaz, Allgaier & Hugenholtz, 2010). In practical terms, there is a high potential of cross contamination throughout the lab, which is particularly problematic in a laboratory where viral work is the main area of research.

In view of the growing interest in the impact of viruses on ecosystems, there is a growing need for bias-limited and preferably easy sequencing. However, the recurring problem on most sequencing platforms is the relative large amount (1 ng to 1 µg depending on the sequencing platform) of DNA needed for the library preparation for sequencing compared to the often low yield of DNA from viral population samples (less than 1 ng). Besides the two options discussed above, a theoretical third option would be to artificially raise the DNA concentration by adding DNA that is naturally non-occurring in any virus. Prime candidate would be the bacterial 16S ribosomal gene as it has not been found in any known virus up to date. In this report, we describe a case study for metagenome sequencing with the Ion Torrent Personal Genome Machine in which we spiked 16S ribosomal DNA to low amounts (0.1, 5 and 7 ng) of DNA from different samples. One of those samples was a bacteriophage population that we compared with traditional MDA amplification.

Material and Methods

To test the technical feasibility of our approach, the protocol was performed three times: once with Enterobacteria Phage P1 (P1-spiked), once with a viral metagenome extracted from an infected bioreactor with a general extraction protocol (DNA-spiked-GP) and finally with a viral metagenome from the same infected bioreactor extracted with a specialised protocol for viral metagenomic DNA (DNA-spiked-SP). The last dataset was also sequenced after MDA amplification.

Sample collection and DNA extraction: general protocol

The bacteriophage population used for sequencing was obtained from a Methylomirabilis oxyfera bioreactor enrichment culture. A sample of 4 L of bioreactor effluent was collected over a period of time of 20 days and was stored at 4 °C. To concentrate the viral particles, the sample was first filtered through a 0.22 µm filter (nucleopore track-etched polycarbonate membrane filters; Whatman, Maidstone, UK) and then further concentrated using 30 kDa Vivaspin Spin Columns (GE Healthcare Life Sciences, Little Chalfont, UK), to a final volume of 3 ml. A P1 reference bacteriophage (genome sequence NC_005856.1) was used as a positive control for DNA extraction. The DNA extraction was performed according to the protocol published by Thurber et al. (2009). Using the Qubit dsDNA HS assay kit (Life), the extracted DNA was quantified at 146 ng/ml.

Sample collection and DNA extraction: specialised protocol

The bacteriophage population used for sequencing was obtained from the same M. oxyfera bioreactor enrichment culture. See Gambelli et al. (2016) for a full description. Bioreactor material was collected over a period of about three months, stored at 4 °C and viral particles were obtained as described before (Gambelli et al., 2016). Briefly, the aggregated microbial biomass was disrupted to free the bacteriophages and viral particles were precipitated using PEG8000 (Guo et al., 2012). Free bacteriophages present in the bioreactor supernatant medium and not within the bacterial aggregates were precipitated by iron chloride flocculation (Cunningham et al., 2015).

The two samples obtained by iron chloride flocculation and PEG 8000 precipitation were pooled together and bacteriophages were concentrated by ultracentrifugation (Optima XE90; Beckman-Coulter, High Wycombe, UK; Rotor: Type 90 Ti; Beckman-Coulter, High Wycombe, UK) at 77,000× g at 4 °C for 1 h. The pellet was resuspended in 1 ml of supernatant and the total DNA was extracted according to the protocol published by Thurber et al. (2009). Using the Qubit dsDNA HS assay kit (Thermo Scientific, Waltham, MA, USA), the extracted DNA was quantified at 0.2 ng DNA.

Sample and library preparation

16S rRNA amplicon PCR

The amplicons of the 16S rRNA gene from Methylacidiphilum fumariolicum strain SolV were obtained by PCR of an in-house sample of isolated DNA from M. fumariolicum (Genbank NZ_LM997411) using primers 616F (5′-AGA GTT TGA TYM TGG CTC -′3) and 630R (5′-CAKAAAGGAGGTGATCC-′3) with the following settings: 5 min at 96 °C, followed by 35 cycles of 40 s at 96 °C, 40 s at 49 °C and 1 min at 72 °C and finalised with an elongation step of 5 min at 72 °C. The final concentration was measured using the Nanodrop ND-1000 (Isogen, De Meern, The Netherlands).

To reduce the risk of sequencing aspecific PCR-products from the PCR-reaction on genomic DNA, amplicons of the 16S rRNA gene of  “Candidatus Kuenenia stuttgartiensis” (GenBank CT573071) were obtained by PCR of an in-house 16S rRNA gene clone using primers pla46 (5′-GGATTAGGCATGCAAGTC-′3) and 630R (5′-CAKAAAGGAGGTGATCC-′3) with the following settings: 5 min at 94 °C, followed by 35 cycles of 40 s at 96 °C, 40 s at 49 °C and 1 min at 72 °C and finalised with an elongation step of 5 min at 72 °C. After amplification, the sample was purified from non 16S ribosomal DNA by excision and re-extraction of the DNA from a 0.9% gel (v/w) using the GeneJET gel extraction kit (Thermo Scientific, Waltham, MA, USA) according to manufacturer’s protocol. The final concentration was measured using the Qubit dsDNA HS assay kit (Thermo Scientific, Waltham, MA, USA).

Enterobacteria Phage P1

To sequence the P1 phage, 5.1 ng DNA from P1 phage (NC_005856.1) was added to 130 ng of amplified 16S rRNA gene DNA from M. fumariolicum strain SolV PCR (referred as P1-spiked sample) and sheared using the Bioruptor® Standard (Diagenode Liege, Belgium) for 10 cycles (1 min on, 1 min off) and prepared according to manufacturer’s protocol (IonXpress Plus gDNA fragment library preparation Rev C.0; Life Technologies, Carlsbad, CA, USA).

Infected bioreactor: general protocol

Approximately 7 ng of viral metagenome DNA was spiked with ∼50 ng of amplified 16S rRNA gene DNA from “Ca. K. stuttgartiensis” (referred as DNA-spiked-GP sample) and sheared using the Bioruptor for six cycles (1 min on, 1 min off) and prepared according to manufacturer’s protocol (IonXpress Plus gDNA fragment library preparation Rev C.0; Life Technologies, Carlsbad, CA, USA).

Infected bioreactor: specialised protocol

Approximately 0.1 ng of viral metagenome DNA was spiked with ∼43.5 ng of amplified 16S rRNA gene DNA from “Ca. K. stuttgartiensis” (referred as DNA-spiked-SP sample) and sheared using the Bioruptor for 6 cycles (1 min on, 1 min off) and prepared according to manufacturer’s protocol (IonXpress Plus gDNA fragment library preparation Rev C.0; Life Technologies, Carlsbad, CA, USA).

Negative control of amplified 16S ribosomal DNA

Approximately 43.5 ng of amplified 16S rRNA gene DNA from “Ca. K. stuttgartiensis” (referred as neg-16S sample), was sheared using the Bioruptor for six cycles (1 min on, 1 min off) and prepared according to manufacturer’s protocol (IonXpress Plus gDNA fragment library preparation Rev C.0; Life Technologies, Carlsbad, CA, USA).

MDA amplification

Approximately 0.1 ng of DNA from the specialised isolation protocol (i.e., DNA-Spiked-SP) was amplified using the Illustra GenomePhi HY DNA amplification kit (GE Healthcare, Piscataway, NJ, USA) as per manufacturer’s protocol. The first amplification round yielded 15 ng of DNA (referred as 1 × MDA sample), Afterwards 10 ng of DNA was used for a second amplification round. This resulted in a yield of 5.4 µg DNA (referred as 2 × MDA sample).

For the P1 Phage, 10 ng of DNA was used for amplification (referred as P1-MDA). A negative control was performed using DEPC water instead of DNA (referred to as neg-MDA), resulting in 6.6 ng of DNA. All DNA was cleaned afterwards using the GeneJET plasmid Miniprep kit (Fermentas, Amherst, MA, USA) according to manufacturer’s protocol, except for step one, in which 200 µl of DEPC was used instead of lysis-buffer.

Starting quantities of DNA and shearing times used for sequence library preparation are given in Table 1. After shearing, the samples were cleaned using a 1:1 volume ratio with AMPure XP beads (Beckman Coultier, High Wycombe, UK) and further prepared for sequencing as per manufacturer’s protocol (IonXpress Plus gDNA fragment library preparation Rev C.0; Life Technologies, Carlsbad, CA, USA).

Table 1 Overview of the parameters and results from sequencing, trimming and mapping to the six viral sequences and P1 phage.

Samplea	DNA (ng)b	16S (ng)b	Shearing cycle; 1 min on/ 1 min off	# reads	Trimming settingsc	# Trimmed reads	Mapped reads to 16Sd	Remaining readse	Expected % of non 16S reads	Observed % non 16S reads	% mapped to #1–6e	% mapped to P1e	
neg-16S	0	43.5	6×	58,989	25–375 bp	58,616	58,293	323	0.00	0.55	0.00e	0,00e	
P1-spiked	5	130	10×	543,649	25–400	522,819	496,250	26,569	3.85	5.08	0.02e	31.82e	
P1-MDA	100	NA	6×	373,351	<25 bp; 25–325 bp	327,451	36	327,415	100	99.99	0.00	94.83	
DNA-spiked-GP	7	50	6×	4,760,807	25–400 bp; 25–350 bp	4,636,949	4,557,290	79,659	14.00	1.72	31.31f	0.01f	
DNA-spiked-SP	0.1	43.5	6×	4,334,460	25–375 bp	4,268,134	3,727,603	540,531	0.23	12.66	64.43f	0.53f	
1 × MDA	5.1	NA	6×	797,971	25–325 bp; 25–400 bp and 15 bp on 5′ end	770,366	3,811	766,555	∼99.9	99.51	1.34	0.15	
2 × MDA	100	NA	6×	190,509	25–375 bp	187,178	1,667	185,511	∼99.9	99.11	1.24	0.09	
neg-MDA	100	NA	9×	94,905	25–340 bp	93,407	775	92,632	100	99.17	0.00	0.00	
Notes.

a P1 refers to phage P1; GP refers to DNA isolated using the general protocol; SP refers to DNA isolated using the specialized protocol.

b Obtained from Qubit measurements.

c Quality trimming = 0.05, ambiguous nucleotide limit = 2.

d Mapping settings = local; 0.5 length; 0.9 similarity.

e Mapping settings = local; 0.5 length; 0.95 similarity.

f Only remaining reads used.

NA not applicable

Sequencing

All samples were sequenced using the Personal Genome Machine Ion Torrent (Thermo Scientific, Waltham, MA, USA) as per manufacturer’s protocol. 1× MDA was sequenced twice, once on a 314v2 chip and once on a 318v2 chip. All other samples were run on a 318v2 chip. All samples were prepared using the Ion PGM™ Sequencing 400 Kit and Ion PGM™ Template OT2 400 kit and sequenced with 850 flow cycles.

Bioinformatics

Trimming and mapping to the SILVA database

After sequencing all samples were trimmed with quality setting of 0.05 and mismatch of 2 using CLC genomics workbench v. 8 (CLCbio, Aarhus, Denmark). Size trimming was dependent on the dataset and values are given in Table 1. To determine the amount of spiked DNA, the reads from each sample were mapped against the SILVA database 16S rRNA v128 (Yarza et al., 2008) (length 0.5, similarity 0.9).

Mapping of P1

Trimmed reads from P1-spiked and P1-MDA were mapped against the genome of P1 phage (NC_005856.1) (length 0.5, similarity 0.95) using CLC genomics workbench v. 8 (CLCbio, Aarhus, Denmark).

Case study of infected bioreactor

To remove genomic DNA from the most abundant microorganism in the bioreactor, the trimmed reads from the DNA-spiked sample were mapped against the genome of “Candidatus Methylomirabilis oxyfera” (Ettwig et al., 2010) (length 0.5, similarity 0.85) and the unmapped reads were assembled (word size 22, bubble size 276, contig length 400), using CLC genomics workbench v. 8 (CLCbio, Aarhus, Denmark) (Fig. 1).

Figure 1 Overview of the methods used to compare MDA and 16S ribosomal DNA spiking.

After DNA isolation, half of the sample was amplified using MDA, while the other half was spiked with 16S ribosomal DNA. Contigs and reads containing 16S ribosomal DNA were filtered out through mapping to the Silva database.

The contigs obtained were subsequently mapped against the SILVA database 16S rRNA v119 (length 0.5, similarity 0.7) and contigs that mapped to “Ca. K. stuttgartiensis” were removed from the database, resulting in 4,088 remaining contigs. These contigs were checked with ESOM (Ultsch & Moerchen, 2005) (default settings) and from this, seven clustering contigs with a high depth were obtained and reassembled with SPAdes (v.3.5.0) (Bankevich et al., 2012) using the ‘trusted-contigs’ and ‘careful’ settings for those seven contigs.

The reads that were used in this SPAdes re-assembly, were obtained by mapping the complete trimmed spiked DNA dataset to the SILVA database 16S rRNA v119 (length 0.5, similarity 0.7). The reads that did not map were used.

Reassembly with SPAdes created 2,094 contigs, 14 of which were bigger than 5,000 bp. From this set of contigs five putative viral genomes of over 15 kpb could be extracted (197 kbp, 86 kbp, 71 kbp, 41 kbp and 17 kbp).

Assembly of the combined 1 × MDA and 2 × MDA datasets (word size 21, bubble size, 265 contig length 1.500) resulted in 689 contigs, ranging from 130,897 to 1,504 nucleotides. With the use of ESOM, one putative viral genome over 15 kb was identified (42 kbp).

GC content

The plots of the GC-content were created with CLC genomics workbench v. 8.

Peak mixture identification

Statistical analysis program R (R Development Core Team, 2008; package mixdist; Macdonald, 2012) was used to identify the different peaks within DNA-spiked-SP, 1 × MDA and 2 × MDA (Supplemental Information 1).

Differential coverage

For differential coverage, reads from the DNA-spiked sample were mapped against the SILVA database 16S rRNA v119 (length 0.5, similarity 0.7). Unmapped reads were combined with the reads from the second 1 × MDA run and assembled (word 35, bubble size 271, contig length 1.000) with CLC genomics workbench v. 8. Subsequent mapping of each read set (length 0.5, similarity 0.8) was performed against the assembled contigs. The depth of both sets was plotted against one another.

Horizontal coverage

To assess how much of each virus was present in each set, the trimmed reads from the three sets were mapped against the six putative viral genomes (length 0.5, similarity 0.95). The number of mapped reads and total length was normalised to the size of the dataset and the length of virus, respectively.

Taxonomic analysis

For an indication of the taxonomic change in the samples, trimmed reads from DNA-spiked-SP, 1 × MDA and 2 × MDA were mapped to the SILVA database 16S rRNA v128. To reduce reads that map solely to the conserved regions of the 16S gene, reads shorter than 75 bp for each set were removed before mapping. Reads from DNA-spiked-SP were mapped strictly to reduce false positives (length 0.6, similarity 0.99). Reads that mapped equally well to multiple references were ignored as well as the reads that matched to Planctomycetes, Verrucomicrobia or Chlamydia. Verrucomicrobia and Chlamydia are in the same PVC super-phylum with Planctomycetes and are likely false positives.

Reads from 1 × MDA and 2 × MDA were trimmed less stringent (length 0.5, similarity 0.95), since these samples are not spiked with 16S ribosomal DNA. Reads that mapped equally well to multiple references were also ignored.

Results

DNA from three different origins was sequenced after being spiked with DNA from two different 16S rRNA gene sources. The number of reads before and after trimming is outlined in Table 1, as well as the number of reads after mapping to the Silva 16S rRNA database.

For the DNA-spiked-GP sample the number of reads was lower than expected (an observed percentage of 1.72% to an expected percentage of 14.0%). The number of reads for the P1-spiked sample was slightly higher than expected (5.08% observed to 3.85% expected). The number of reads for the DNA-spiked-SP sample was 55 times higher than expected (12.66% observed to 0.23% expected).

Mapping and assembly of P1

Reads from P1-spiked and P1-MDA samples were both mapped to the genome of P1 phage (NC_005856.1) and the results are overall the same regarding horizontal coverage of the genome. A drop in coverage from basepair 5,710 to 15,006 for both samples is clearly visible, although in both samples reads do still map to the genome (Fig. 2).

Figure 2 Coverage of the Enterobacteria P1 phage after sequencing using MDA amplification and 16S ribosomal DNA spiking.

Case study of an infected bioreactor Viral DNA extracted from biomass from an infected bioreactor was sequenced following two different approaches: MDA amplification (two samples) and non-amplified DNA spiked with bacterial 16S rRNA gene DNA. This resulted in three datasets comprising of a total of 770,366 trimmed reads for 1 × MDA and 187,178 trimmed reads for 2 × MDA. After the reads from the non-amplified spiked DNA were trimmed and mapped against the SILVA database, the final number of reads left was 529,481. After trimming, GC graphs were created showing the GC distribution of each dataset (Fig. 3) and underlying peaks were identified using the mixdist package from R (Supplemental Information 1). In the DNA-spiked-SP sample a total four peaks were identified (62%, 57%, 43% and 42% GC), with peaks at 62% GC and 57% GC as the two most prominent ones (respectively 0.37 and 0.47 out of 1). After one round of amplification, this has shifted to 57% GC (0.67) and 45% GC (0.24). The peak at 62% completely disappeared. After two rounds of amplification, the peak at 57% GC has been further reduced to 0.08, while both peaks around 43% GC (0.69) and 42% GC (0.23) have increased.

Figure 3 Distribution of reads obtained from Ion Torrent sequencing using three different sample preparation methods based on their individual GC content in % of the total number of reads in one sample.

Figure 4 Comparison of the viral reads from the individual datasets mapping to the six assembled virus genomes (Green, DNA-spiked sample; Blue, 1 × MDA; Red, 2 × MDA).

(A) Horizontal coverage of the viral genomes with reads from the individual datasets. (B) Distribution of the reads from the individual datasets over the assembled viral genomes. (C) Depth (vertical coverage) of the viral genomes with reads from the individual datasets as a measure of abundance.

A total of six different viral sequences (putative genomes) could be assembled using the reads from datasets 1 × MDA, 2 × MDA and DNA-spiked-SP by a combination of various methods (see Materials & methods and Gambelli et al., 2016) for a more elaborate description of the viruses). The DNA-spiking method resulted in five complete viral genomes over 15 kbp in size while from the MDA set only one complete viral genome could be recovered after assembly. The length of the five viral genomes extracted from the spiked dataset ranged from 197 kbp to 17 kbp with GC contents from 67% to 54%. The length of the viral genome extracted from the MDA dataset is 42 kbp with a GC content of 35%. Five of the viral genomes (197 kbp, 86 kbp, 71 kbp, 42 kbp and 17 kbp) contained the same sequence on each end of the contig indicating a full circular genome. Figure 4 shows the comparison of the total amount of virus genomes that could be retrieved from the individual datasets. Figure 4A shows that all six virus genomes could be recovered in nearly complete length from the DNA-spiked dataset. In contrast, MDA amplification clearly lowered these percentages. Figure 4B shows the percentage of reads from each viral metagenome mapping to the six virus genomes. With the MDA amplified samples, the percentages of reads mapping is low with a comparable relative abundance. The non-amplified sample not only shows more variation in relative abundance, but the total amount of mapped reads comprises over half of the original dataset. With the MDA method, the number of mapped reads drops to lower than 3%. Figure 4C shows the average depth of each viral genome after mapping the reads of the individual datasets. The data are comparable to Fig. 4B. Using the DNA-spiked dataset, high depth (>35) is found for five out of six viral genomes. The MDA datasets show very low depth and only gave better results for the 42 kbp virus.

Assembly of the reads from the DNA-spiked sample and the 1 × MDA sample resulted in a total of 1,644 contigs. Differential coverage of these contigs is shown in Fig. 5. From the figure it is clear that the depth for each set of contigs differs completely since two similar sets would result in contigs placed on a diagonal straight line. However, the figure shows an almost perpendicular distribution. Whereas the contigs originating from the DNA-spiked sample are placed in a low horizontal line, the contigs originating from the 1 × MDA sample are vertically placed on the coverage graph. When looking at the location of contigs, it becomes clear that high GC% contigs (green) are present within the DNA-spiked dataset but are low in the amplified sample while low GC% contigs (pink) are much more abundant in the 1 × MDA dataset. The figure also demonstrates the wide range in sequencing depth when no amplification was applied (maximum around 380) in contrast to amplification, as the sample depth was lowered tenfold, with a maximum around 38.

Figure 5 Differential coverage of the viral contigs assembled using a combination of the DNA-spiked sample and the 1 × MDA sample with each individual read sets.

Each circle represents a contig present after assembly and the placement in the plot shows the abundance of the contig for each read set. Two similar read sets would result in a diagonal straight line. GC content of the different contigs is indicated as depicted in the colour scale and the size of the bubble depicts the length of the contig. Three outliers caused by the MDA amplification method are not shown in the plot.

Table 2 Overview of the 16S reads in DNA-spiked-SP, 1× MDA and 2× MDA in number of reads and percentage.

The percentages are colour-coded in a gradient from 0% (red) to 100% (green). The reads were identified by mapping to the SILVA 16S rRNA database v128. DNA-spiked-SP (length 0.6, similarity 0.99):1 × MDA and 2 × MDA (length 0.5, similarity 0.95). Ambiguous reads were removed from the set.

	DNA-spiked-SP	1 × MDA	2 ×MDA	
	#	%	#	%	#	%	
Rhizobiales	61	18.9	1	1.1	0	0.0	
OPB41	2	0.6	0	0.0	0	0.0	
Actinomycetales	3	0.9	0	0.0	0	0.0	
SAR11	6	1.9	5	5.4	1	3.0	
Latescibacteria	7	2.2	0	0.0	0	0.0	
Clostridia	25	7.8	0	0.0	0	0.0	
Saccharibacteria	7	2.2	4	4.3	4	12.1	
Omnitrophica	18	5.6	0	0.0	0	0.0	
Streptomycales	12	3.7		0.0	0	0.0	
Microgenomates	2	0.6	17	18.5	6	18.2	
WS6	0	0.0	53	57.6	15	45.5	
Woesearchaeota__DHVEG-6	0	0.0	1	1.1	4	12.1	
Parcubacteria	1	0.3	1	1.1	0	0.0	
Lactobacillales	0	0.0	2	2.2	0	0.0	
Staphylococcaceae	1	0.3	2	2.2	0	0.0	
Bacillales	0	0.0	5	5.4	0	0.0	
Chloroflexi	15	4.7	0	0.0	0	0.0	
Pseudomonadales	0	0.0	0	0.0	1	3.0	
Thaumarchaeota	0	0.0	0	0.0	1	3.0	
Nitrospira	13	4.0	0	0.0	0	0.0	
Actinobacteria	73	22.7	0	0.0	0	0.0	
Nitrospina	17	5.3	0	0.0	0	0.0	
Cyanobacteria	0	0.0	1	1.1	1	3.0	
Other	59	18.3	0	0.0	0	0.0	
Total read count	322	100.0	92	100	33	100	

For the taxonomy, mapping of the reads to the Silva database resulted in 322 reads for DNA-spiked-SP. Groups represented by 4% of the reads or more are Actinobacteria (22.7%), Rhizobiales (18%), Clostridia (7.8%), Omnitrophica (5.6%), Nitrospina (5.3%), Chloroflexi (4.7%) and Nitrospira (4.0%) (Table 2). The remaining reads are 18.3% of the total. For 1 × MDA 92 reads were mapped against WS6 (57.6%), Microgenomata (18.5%), SAR11 (5.4%), Bacillales (5.4%) and Saccharibacteria (4.7%). For 2 × MDA 33 reads were mapped against WS6 (45.5%), Microgenomata (18.2%), Saccharibacteria (12.1%) and Woesearchaeota_DHVEG-6 (12.1%).

Discussion

Here we describe a case study in which we sequenced low amount of DNA (viral DNA) after spiking 16S rRNA gene DNA and compare this to sequencing after traditional MDA amplification.

For both metagenomes containing viral genomes up to 200 kbp and the single P1 genome of 95 kbp it was possible to retrieve the original data after spiking with 16S rRNA gene DNA. For the DNA-spiked-GP, no MDA data were available for an in-depth comparison. However after mapping reads to the genomes, it was clear that viruses 1 to 6 were present. The distribution was different from the DNA-spiked-SP sample, which can either be caused by bias introduced by the extraction protocol or a change in the viral population.

Comparing P1-spiked and P1-MDA by mapping back their reads to the P1 genome resulted in different depth values, being 4,438 for P1-MDA and 65 for P1-spiked, respectively. The horizontal coverage for both sets was the same. This is not unexpected, since there is no preferential genome for the MDA enzyme to amplify. Curiously, the P1 phage used during this experiment seems to have evolved during the time in our lab, as in both the MDA and spiking method a gap appeared at the same locus. However, the depth did not drop to 0 in both samples, indicating that at least a small portion of the original viral population was still present.

Two controls were sequenced along with the samples (Table 1). One control was the 16S rRNA gene DNA without sample DNA. Although no other reads were expected, 0.55% of the reads was not of 16S origin. Since most of those reads matched to M. fumariolicum strain SolV in a BLAST search, the most likely explanation is contamination during library preparation. The level of contamination is not high enough to account for the amount of non-16S in the spiked samples. On top of that, the composition of the non-16S reads is different for the negative control and samples.

The second control consists of a MDA run performed on DEPC-water. In absence of template DNA, MDA still produces dsDNA, since at some point during amplification, the random hexamers act like a template instead of primer. Moreover, MDA is by design very sensitive to small amounts of DNA, therefore any contamination is easily picked up. The contamination however did not consist of the sampled viral or bacterial community.

Differential binning of spiked DNA and the 1 × MDA not only shows a remarkable difference between both samples in terms of composition, but also demonstrates the loss of information about the abundance of contigs in the samples. Depth (vertical coverage) of a contig is a measurement of abundance of that contig in a sample. In the DNA-spiked sample, the maximum depth is 380. In the MDA amplified sample the maximum depth has dropped to 38. Moreover, the best covered contigs after MDA amplification are not the contigs best covered in the spiked DNA. This shift means that sequencing depth after MDA treatment is a poor indicator of abundance of contigs in the original sample. This also complicates binning afterwards, as depth or abundance is often used as a parameter in the binning of metagenomes.

Like in previous reports (Kim & Bae, 2011; Marine et al., 2014), our experiments show the bias that MDA amplification of metavirome or metagenome DNA introduces into the dataset. In the extreme case of applying MDA twice, the dataset is completely changed with major consequences for results obtained, which as such may not reflect the sampled ecosystem. Although taxonomic analysis is troublesome on a sample that is spiked with short 16S ribosomal DNA reads because of the conserved regions, a taxonomic change was observed when analysing the residual 16S genes that were present as contamination in the extracted viral DNA (Table 2). The DNA-spiked-SP showed a variety of organisms present with a few groups in high numbers like Rhizobiales and Actinobacteria. The 1 × MDA and 2 × MDA show a shift towards the WS6 group and the group of Microgenomata. Analysis on a sample that is spiked with 16S rRNA gene DNA is not straightforward because of the risk of false positives that are likely to occur. Nonetheless, the change (from 18.9% to 1.1% for Rhizobiales and 22.7% to 0% for Nitrospira) was so pronounced that false positives are very unlikely the only cause of the shift. Moreover the shift towards 1 × MDA (from 0.6% to 18% and the group Microgenomata and from 0% to 57.6% for the WS6) would be smaller with false positives.

Several different reasons have been given in literature regarding the bias of MDA such as fragment length, GC-content, quality of DNA and even different MDA kits (Abulencia et al., 2006; Kim & Bae, 2011; Yan et al., 2004; Yilmaz, Allgaier & Hugenholtz, 2010). Although there seems to be no general consensus, for our results it seems that specifically DNA with a low GC content is favoured. This may lead to a considerable shift of overall GC content causing a severe underestimation of the quantity of viruses with a higher GC percentage and they might thus be easily overlooked in metagenomic research.

From the spiked datasets, it is clear that the final ratio of the sample versus 16S rRNA gene DNA can vary considerably. Possible explanations could be differences in DNA concentration measurements or adjustments made for an optimal sequencing run prior to the final shearing of the sample (Supplemental Information 2). The amount of reported 16S rRNA gene reads can also be influenced by the mapping settings (stringent vs not stringent or local vs global). In practical terms, it is easier to assemble the complete dataset and filter out 16S rRNA gene sequences afterwards. The 16S ribosomal fragments would be longer and fewer in numbers after assembly and thus easier to identify. However, pre-filtering most of the 16S rRNA gene reads will speed up assembly.

A positive side effect of the spiking protocol is that it lowers the risk of contamination. By choosing the 16S rRNA gene of a microorganism which is very uncommon to the laboratory, contamination can easily be recognized and filtered out. One could even use the eukaryotic 18S rRNA gene in a prokaryotic-based lab and vice versa.

In this case study we described a protocol for viral metagenome sequencing of extremely low amounts of DNA (as low as 0.1 ng) that is unbiased, inexpensive, easy and readily available for any lab with sequencing facilities and that can possibly be extended for other non-16S/18S containing DNA like plasmids, or other Next Generation Sequencing platforms like MinIon or Illumina.

Conclusions

When dealing with low quantities of DNA for Next Generation Sequencing from environmental samples like water or forensic samples, multiple displacement amplification (MDA) of the DNA from the sample might not be the method of choice to obtain enough starting material. In a case study we have shown that spiking with 16S rRNA gene DNA could be an alternative, eliminating the need for amplification. The reads resembling the added DNA can be easily discarded afterwards and we observed a very low bias in the dataset compared to the MDA method.

Supplemental Information

Supplemental Information 1 Peak identification using R-package mixdist

For each sample is shown from top to bottom: the graphs, final peak statistics and R-code. The statistics show the mean GC content for a peak (mu), the SD of the peak (sigma) and proportion of each peak of the total graph (pi). In the graph, red triangles depicts mu, the red lines are the individuals peaks, the green line is the combined peak and in blue is the original histogram.

Click here for additional data file.

Supplemental Information 2 Shearing patterns during Library preparation

Pre-shearing for Spiked-DNA-GP: 5 microliter was chosen because there was little difference between 5 and 6 microliter at 400 bp, although the shearing of 6 microliter was better. However 5 microliter means less 16S ribosomal DNA and more DNA from the actual sample. (top graph and table).

Click here for additional data file.

We thank Simon Guerrero for supplying the bioreactor material.

Additional Information and Declarations

Competing Interests

Author Contributions

DNA Deposition

The authors declare there are no competing interests.

Geert Cremers conceived and designed the experiments, performed the experiments, analyzed the data, contributed reagents/materials/analysis tools, wrote the paper, prepared figures and/or tables.

Lavinia Gambelli performed the experiments, contributed reagents/materials/analysis tools, reviewed drafts of the paper.

Theo van Alen conceived and designed the experiments, performed the experiments, contributed reagents/materials/analysis tools, reviewed drafts of the paper.

Laura van Niftrik analyzed the data, reviewed drafts of the paper.

Huub J.M. Op den Camp conceived and designed the experiments, analyzed the data, wrote the paper, prepared figures and/or tables.

The following information was supplied regarding the deposition of DNA sequences:

The raw sequence data were submitted to the European Nucleotide Archive and are available under accession number PRJEB20134.

The viral genomes are available from GenBank under accession numbers, KX853510, KX853511, KX853512, KX853513, KX853514.

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
