# Peer review of "Bioreactor virome metagenomics sequencing using DNA spike-ins"

_PeerJ, doi:10.7717/peerj.4351_

## Round 0.1 · original submission · Major Revisions

· Academic Editor

Major Revisions

Thank you for your interesting submission to PeerJ. As you can see, the three reviewers have carefully reviewed your manuscript and think the method you have described is potentially of interest to those sequencing viral metagenomes.

However, each reviewer raise concerns about the manuscript as currently presented. I would like you to consider each point they have raised carefully in your response.

As this is a methods paper it is important that a reader can determine whether the method would be helpful for them. It is therefore important to demonstrate that the method is repeatable, and the range of concentrations that it may work for. Therefore it is important that replicates are provided - I recommend three of each condition. The manuscript would also be strengthened by the inclusion of further controls (high input DNA and negative water), and potentially a range of spike-in DNA amounts.

In light of the need for additional - in my view critical - additional experiments I have rendered a decision of Major Revisions and look forward to receiving your response.

·

Basic reporting

- Minor points:
o A clearer picture of the potential samples that would be processed with this protocol should be presented. The assumption is that these would be environmental samples, but even that is too broad. Water, soil, plant material, clinical samples?
o Lines 42 and 62: “low quantity” and “low yield” of DNA. This is too vague. The reader should also be presented with DNA concentration ranges from environmental samples so that they can fully understand the problem. Obviously some would contain enough DNA.
o Line 61: “relative large amount of DNA needed”: Again, this is too vague. Please present the input requirements for several popular library prep kits so that the readers can clearly see how the expected DNA extraction yield compares to the kit requirements.
o Relating to the last point, with several commercial kits, DNA can be prepared for sequencing using < 1ng of DNA. For example, even though Nextera XT is optimized for at least 1 ng of DNA, as little as 5-10 pg can be used to generate libraries. However, sequencing from pg of DNA opposed to ng could also skew the viral populations. It would be nice to discuss this approach as an alternative.

Experimental design

- The authors present a potential solution to a problem that many sequencing labs face, low input concentrations. However, the experiments and data presented here are not thoroughly convincing. Major points:
o There are no controls. A high input (i.e. 1 or more ng DNA) control would provide the expected results. At this point, I cannot tell if there should only be 6 viruses present in the population or more. Also, a water or negative control would help to convince me that some of these reads were not contamination. A comparison between spiked and MDA is not enough.
o N=1 for all test groups. This should be expanded to at least n=3 for all groups for your results to be sufficiently convincing.
o Only 16S reads were removed the DNA-spiked reads (via mapping to SILVA). All groups should be treated to the same analysis. This may have also removed real 16S reads that were present in the bacteriophage population, but the 16S were left in the MDA samples. This may have accounted for some of your % reads aligning to the viral sequences.
o A dilution series of input DNA would be required to justify how much input DNA is required for this protocol. Likewise, why was 43.2 ng chosen as a spike-in concentration? Adding a test range of spike-in amounts, in addition to the dilution series of input concentrations, would help potential users of this protocol know what combinations to use.

- Minor points:
o Include a table that shows the data per library. My suggestions:
♣ total number of reads pre-trimming
♣ total number of reads post quality trimming
♣ total number of reads post 16S removal
♣ Percent processed reads aligning to bacteria and viruses #1-6
o Line 218: show the data shift towards bacteria genes. This could be included in the table.
o An overview figure of the method would help readers better understand the process.

Validity of the findings

o Line 236: Conclusion that you could examine the viral metagenome using an extremely low input (0.1 ng DNA needed) is not fully supported here. See experimental design. Controls and additional comparisons are required.

Comments for the author

This very well could be a really neat, easy solution to low input amounts of DNA heading into the library prep process. The data shown supports the authors method, but I am hesitant to be convinced until I see additional replicates and controls. Also, the input and spike-in ranges need to be defined.

Reviewer 2 ·

Basic reporting

The paper meets the general requirements for this area.

Experimental design

One major issue for the study is that the comparison is made with N=1. The reproducibility of the spike-in library data is unknown. Since this paper is arguing for the virtue of the method, at a minimum, the authors must repeat the library construction and sequencing of the spike-in sample, and define the reproducibility of the results.

Ideally, I would prefer to have seen a defined viral metagenome (mixture of purified, known viral genomes) assessed by the spike-in vs MDA methods. This would avoid issues arising from uncertainties as to the composition of the unknown viral metagenome used in this experiment (e.g. the presence of small bacteria that pass through the viral purification step that the authors mention, etc). Nonetheless, useful data can certainly be extracted from the authors approach.

The mass ratio of the spike in—0.1 ng with 42 ng of spike would suggest an ~ 400:1 ratio of corresponding reads. What are the relative numbers of reads that are derived from the spike to non-spike sequences? If they do not approximate 400:1, the authors should discuss possible explanations for the actual ratio. The authors should present statistics regarding the actual number of reads at each step (total, mapped to 16S, etc) and not just the relative abundances and %. By my ballpark calculations from Figure 3 (197k genome * 300x coverage)=60 MB of viral sequence. This constitutes ~40% of the total viral metagneome, so at least 120Mb out of the total run is from the viral metagenome. Seems like the actual ratio has to be much lower than 400:1. Why would this be?

The authors claim that 6 putative viral contigs were obtained. However, no information is presented regarding their identity—what viruses are they most similar to? What criteria were used to call them (and not any of the many other contigs obtained) viral? More detail needs to be provided about this.
Also, it is not clear what is meant in line 182 “The 
DNA-spiking method resulted in five viral genomes while the MDA set only resulted in one.” The authors demonstrate that reads map to all 6 of these viruses from both methods—so is it the assemblies of the two that differ and the criteria for obtaining a viral candidate? This needs clarification.

What is unique about the 42k contig that is it is favored in the MDA dataset?

Figure 3 is a bit complicated. Most importantly, the authors depict circles of different sizes in the graph, but provide no explanation in the legend or the text as to what the diameter of a given data point reflects. Is this the size of the contig?

Validity of the findings

GC content shift is a simple indicator that there has been a change, and the authors’ data clearly demonstrate a change in the GC content of the reads with MDA. However, the paper would be strengthened if the authors were to analyze the taxonomic distribution of the reads generated by spike vs MDA to explicitly define which taxa are increased/decreased in representation by the MDA process.
Given that one known bias is that MDA will preferentially amplify small circular genomes, is it possible that there is some low GC-content small circular DNA template contaminant (e.g. plasmids or dark matter circular viruses) that is being amplified and driving this observed increase in low GC content? Because the authors conclude from this single example that MDA biases against high GC and favors low GC, it is important to determine as best as possible the source(s) that contribute to their observations. Again, this is a single sample—would other samples show this same pattern? The authors should temper their conclusions in the absence of additional data.

·

Basic reporting

The authors report here a simple but efficient method to sequence samples with low amount of DNA such as is often obtained in viral metagenomic studies. The introduction and material & methods are overall clear and well structured and below are some minor comments. However, the authors should globally improve the quality of the report in the results and the discussion for both english and meaning/clarity (also find some additional more specific comments in further review sections below).

Introduction:
L35-38: Please cite some relevant research and reviews for the definition of the viral "black matter" and its development with the advent of NGS.
M &M:
There are several repetitions in lines L138-144 and L145-150. Please clarify the difference between those lines.
Results:
L182 The M&M states "one more putative viral genome", which implies that the other genomes were also retrieved, which is contradicted here. Please revise the sentences.
L202-203 What is "a perpendicular distribution"? Please better describe the results obtained.
Discussion:
L215: Replace "is emphasizing" by emphasizes
L238: Replace sequence facilities by sequencing facilities

Experimental design

The authors discuss the annotation of contigs in the discussion section but no information is provided in the materials and methods. They should describe the method used for annotation and to distinguish bacteria vs virus genes.

Validity of the findings

Given the bimodal distribution of GC content in reads (Fig 1), it would be interesting to fit the values with two Gaussian distribution for each method. This would enable the authors to obtain reliable estimate in terms of mean and variance that would better show the shift due to MDA amplification (instead of approximate values).

L219-222 Please extend the discussion on the WS6-like bacteria. Are reads mapping to these bacteria also observed in the spiked DNA sample?

L223: Please discuss more largely this bias in the light of other studies reporting biases in MDA

L227-231: Rewrite this paragraph to clearly describe the differences observed, the implications, and better convey the message on the potential of DNA spiking instead of MDA with regard to the estimation of organisms' abundance.

The conclusion could be made stronger by avoiding vague terms such as "a more valid alternative" or "the obvious bias" (which may not be that obvious - are we talking about GC content bias?) and clearly stating the advantages of DNA spiking over MDA

---

## Round 0.2 · Major Revisions

· Academic Editor

Major Revisions

The three reviewers have returned a steep gradient of opinion on your article. Two reviewers felt quite strongly that the central claim of the paper, that of a new method for low input viral metagenome sequencing, is not supported by the data in the manuscript.

The main issue raised is that of reproducibility of the method. There is currently not enough experimental data, such as use of known mock communities, or ranges of spike-in input, in order to confidently understand how well the method will perform on other samples. These are important considerations for a Methods paper and will limit its relevance to readers wishing to test the method on their own samples.

I do appreciate the difficulty of repeating the method on precious samples. One reviewer felt strongly that a representative mock community virome was an achievable aim, and I agree.

I will accept a further resubmission as the manuscript is interesting as a case study. But there needs to be a significant refocusing of the manuscript away from describing a method, and instead re-focusing the work as a case study about sequencing viromes from a bioreactor.

If you wish to resubmit, please take care to ensure that this change of focus is fully reflected through the manuscript, including a change in title. If this is not done I am afraid I will not be able to accept an updated version. Please also note the requirements to improve the standards of reporting as suggested by Dr. Bertelli.

·

Basic reporting

Table 1 is confusing. P1, SP, and GP need to be defined.

Experimental design

See general comments

Validity of the findings

See general comments

Comments for the author

The authors state several times within the rebuttal that critiques were not within the scope of the current article. Without knowing the limits of input DNA detection and the optimal ratios of spike-in concentrations, this manuscript is essentially just about how the authors rescued viral DNA from their one “precious” bioreactor sample. Thus, if they cannot fulfill these requirements, the title of the manuscript “DNA-spiking in viral metagonome sequencing: A new method with low bias” is misleading. Interested readers would see this expecting a validated method to work for their own samples coming from a range of environments, but instead would find a case study about sequencing DNA from a bioreactor. It is not clear how beneficial this method would be outside of the current design, which drastically limits the impact. If the authors are unwilling to make this change, then perhaps it should be refocused – i.e. change title and remove suggestions that this is a general use protocol.

Reviewer 2 ·

Basic reporting

no comment

Experimental design

There are still no replicates provided in this study. The paper purports to describe a useful method, but there is insufficient evidence of the reproducibility of this method.

The major point of the manuscript is the description of a method (per the title). I fully appreciate that the materials used by the authors are very dilute—however, to demonstrate a method for the first time, those do not seem to be the appropriate materials to use. Rather, it should be straightforward to reconstruct a virome (by mixing defined quantities of known viral genomes) to demonstrate the performance, reproducibility and utility of this method. Subsequently extending this validated approach to one or more true unknown sample (such as the samples used by the authors) would then be convincing.

I do not understand why the authors argue “a defined viral metagenome is not really feasible." "The aim is to describe a method usable for low amounts of DNA obtained from precious samples”.

If the authors’ true interest is to analyze the viromes in their dilute samples, then they should retitle and rewrite the paper to focus on the viruses they were able to detect rather than present the results as a new method.

I do know on what basis the authors assert “It is very unlikely that MDA preferentially amplifies small circular genomes.” The literature is extensive about rolling circle amplification by phi29 polymerase, and that compared to larger, linear fragments, preferential amplification of small circular genomes will occur. See for example:
A sequence-independent strategy for detection and cloning of circular DNA virus genomes by using multiply primed rolling-circle amplification. Rector, A. et al. J. Virol. 2004; 78: 4993–4998
Rolling-circle amplification of viral DNA genomes using phi29 polymerase
Reimar Johne, Hermann Müller, Annabel Rector, Marc van Ranst, Hans Stevens 2009 Trends in Microbiology.

Validity of the findings

see above

·

Basic reporting

The authors have added useful citations of previous work, and corrected some sentences. However, they have also significantly expanded the previous version of the article, and the quality of the reporting is not satisfactory. Different samples have been prepared with different methods, and the results do not always clearly specifiy the sample/protocol used, as for example in Figure 3 legend ("on one sample"). Unnecessary repetitions are often found, for example "Several different causes have been given in literature regarding the bias of MDA. Reasons such as fragment length, GC-content, quality of DNA and even different MDA kits 416 (Abulencia et al., 2006; Kim & Bae, 2011; Yan et al., 2004; Yilmaz et al., 2010) have been implicated in MDA bias.". The authors should improve the overall quality of the report and the quality of English writing, trying to facilitate the reader's understanding as the complexity of the various samples sequenced and methods used has increased.

Experimental design

The software used for the mapping of the reads to P1 phage (L183) as well as to the SILVA database should be mentioned.

Validity of the findings

Results reported in Table 2 have been obtained using different cutoffs to map reads to the SILVA database. As such, the results in column DNA-spiked-SP and 1x-2xMDA are not comparable. Why did the authors change their cutoffs ? The 16S used to spike the sample has been removed prior to the mapping, and therefore should not interfere with this second mapping. The presence of chimeric reads should be checked for before using appropriate tools.

Figure 1 showing the methods used is a good addition to facilitates the reader's understanding. Given its current 2/3 of a page width size, it could be expanded to include as well a representation of the different sample preparation, sequencing (P1, GP, SP, spiking, MDA) & naming convention.

---

## Round 0.3 · Minor Revisions

· Academic Editor

Minor Revisions

Thank you for your resubmission. I approve of the changes you have made which have improved the manuscript. I do not see a reason for it to undergo another round of peer review.

However I would like you to make one further change which is to the title. It should be more specific to reflect that it is a case study of a particular environment. I would suggest "Bioreactor virome metagenomics sequencing using DNA spike-ins".

Please return with this change and I will render the acceptance decision immediately.

---

## Round 0.4 · accepted · Accept

· Academic Editor

Accept

Many thanks for making the title change as requested and I am pleased we were able to accept your article.